# (S)-Reutericyclin: Susceptibility Testing and In Vivo Effect on Murine Fecal Microbiome and Volatile Organic Compounds

**DOI:** 10.3390/ijms22126424

**Published:** 2021-06-15

**Authors:** Bernhard Kienesberger, Beate Obermüller, Georg Singer, Barbara Mittl, Reingard Grabherr, Sigrid Mayrhofer, Stefan Heinl, Vanessa Stadlbauer, Angela Horvath, Wolfram Miekisch, Patricia Fuchs, Ingeborg Klymiuk, Holger Till, Christoph Castellani

**Affiliations:** 1Department of Paediatric and Adolescent Surgery, Medical University of Graz, 8036 Graz, Austria; bernhard.kienesberger@medunigraz.at (B.K.); georg.singer@medunigraz.at (G.S.); barbara.mittl@medunigraz.at (B.M.); holger.till@medunigraz.at (H.T.); christoph.castellani@medunigraz.at (C.C.); 2Department of Biotechnology, University of Natural Resources and Life Sciences Vienna, 1190 Vienna, Austria; reingard.grabherr@boku.ac.at (R.G.); sigrid.mayrhofer@boku.ac.at (S.M.); stefan.heinl@boku.ac.at (S.H.); 3Department of Internal Medicine, Division of Gastroenterology and Hepatology, Medical University of Graz, 8036 Graz, Austria; vanessa.stadlbauer@medunigraz.at (V.S.); angela.horvath@medunigraz.at (A.H.); 4Center of Biomarker Research (CBmed), 8036 Graz, Austria; 5Experimental Research Center, Department of Anesthesiology and Intensive Care, Rostock University Medical Center, 18057 Rostock, Germany; wolfram.miekisch@uni-rostock.de (W.M.); patricia.fuchs@uni-rostock.de (P.F.); 6Gottfried Schatz Research Center, Department of Cell Biology, Histology and Embryology, Medical University of Graz, 8036 Graz, Austria; ingeborg.klymiuk@medunigraz.at

**Keywords:** reutericyclin, bacteriocin, microbiome, antimicrobial activity, VOC, isoform, antibiotic, resistance

## Abstract

We aimed to assess the in vitro antimicrobial activity and the in vivo effect on the murine fecal microbiome and volatile organic compound (VOC) profile of (S)-reutericyclin. The antimicrobial activity of (S)-reutericyclin was tested against *Clostridium difficile*, *Listeria monocytogenes*, *Escherichia coli*, *Enterococcus faecium*, *Staphylococcus aureus*, *Staphylococcus (S.) epidermidis*, *Streptococcus agalactiae*, *Pseudomonas aeruginosa* and *Propionibacterium acnes*. Reutericyclin or water were gavage fed to male BALBc mice for 7 weeks. Thereafter stool samples underwent 16S based microbiome analysis and VOC analysis by gas chromatography mass spectrometry (GC-MS). (S)-reutericyclin inhibited growth *of S. epidermidis* only. Oral (S)-reutericyclin treatment caused a trend towards reduced alpha diversity. Beta diversity was significantly influenced by reutericyclin. Linear discriminant analysis Effect Size (LEfSe) analysis showed an increase of *Streptococcus* and *Muribaculum* as well as a decrease of butyrate producing *Ruminoclostridium*, *Roseburia* and *Eubacterium* in the reutericyclin group. VOC analysis revealed significant increases of pentane and heptane and decreases of 2,3-butanedione and 2-heptanone in reutericyclin animals. The antimicrobial activity of (S)-reutericyclin differs from reports of (R)-reutericyclin with inhibitory effects on a multitude of Gram-positive bacteria reported in the literature. In vivo (S)-reutericyclin treatment led to a microbiome shift towards dysbiosis and distinct alterations of the fecal VOC profile.

## 1. Introduction

The importance of the intestinal microbiome has gained wide scientific interest in both health and disease. A plethora of factors including nutritional habits, medication or chronic illnesses influences the microbiome composition. Many diseases such as type II diabetes, obesity, cancer cachexia, chronic cardiovascular or chronic inflammatory bowel disease have been associated with dysbiosis—an alteration of the composition of the microbiome with negative effects on the host [1,2,3,4,5]. Therefore, research has focused on possibilities to shift the intestinal microbiome towards a “healthier” composition as a possible therapeutic or supportive approach. Apart from dietary modifications, antibiotics or stool transplantations, this could be achieved by nutritional supplementation with either pre-, pro-, syn- or postbiotics. While probiotics are generally considered as safe, infections with probiotics may occur in vulnerable patient cohorts on rare occasions. Therefore, alternatives to live microorganisms to manage dysbiosis are of interest [6].

Some intestinal bacteria are able to produce specific substances inhibiting growth or inactivating other competitive strains in order to defend their own habitat. This bacterial defensive strategy is based on different classes of antimicrobial compounds. Some antimicrobial metabolites like acetic acid, lactic acid, or reuterin are small organic compounds [7,8]. Others like reutericin are short, ribosomally synthesized, post-translationally modified peptides termed bacteriocins [9,10]. Finally, reutericyclin is another form of antimicrobial substance synthesized by a non-ribosomal peptide synthetase [11]. Due to their antimicrobial effects, all these substances, which are also contained in postbiotics, may be attractive alternatives to treat dysbiosis, such as in immunocompromised patients.

In 2000, reutericyclin (4-acetyl-1-[(E)-dec-2-enoyl]-3-hydroxy-2-(2methylpropyl)-2H-pyrrol-5-one) was first mentioned as a low molecular weight antibiotic produced by certain lactic acid bacteria [12]. Reutericyclin, with a molecular mass of 349 Da, is a typical 1,3-bisacyltetramic acid and consists of four methyl and seven methylene groups, as well as two aliphatic and two olefinic methine groups, as shown in the corresponding HSQC spectrum [13,14]. With this structure, reutericyclin is an amphiphilic molecule consisting of a hydrophilic, negatively charged head group with two hydrophobic side chains [15]. It is a weak acid and its activity is increased at low pH levels [14]. Reutericyclin exists in (R)- and (S)-isoforms at C_5_. Reutericyclin was found to inhibit growth of many Gram-positive bacteria [16]. This bacteriocidal effect of reutericyclin has been attributed to its action as a proton ionophore [15]. In this way, it selectively dissipates the bacterial transmembrane potential [17,18]. Fungi, yeasts and gram-negative bacteria, however, have been described as being resistant to reutericyclin [15]. Reutericyclin producing strains of *Limosilactobacillus (L.) reuteri* are broadly utilized in food fermentations [12] and have also been previously described as a stable constituent of the intestinal microbiota of both humans and animals [19]. Therefore, the application of reutericyclin appears to be a save approach. However, at present there are only limited reports regarding the effect of reutericyclin on the intestinal microbiome in vivo. While there is no such study on reutericyclin as pure substance, there are a few reports investigating the effect of reutericyclin producing probiotic *L. reuteri* in piglets [20,21]. In one study, this decreased the abundance of the *L. reuteri* group in fecal samples in comparison to controls [21]. These results suggest that reutericyclin is a subtle but significant modulator of the Lactobacillus community in pigs [21]. In contrast, Wang et al. report that the probiotic derived antimicrobial compound reutericyclin had only a limited impact on swine intestinal microbiota [20].

The aims of this study were to examine (1) the in vitro antimicrobial activity of (S)-reutericyclin, (2) its in vivo effect as pure substance on the fecal microbiome, and (3) the fecal volatile organic compound (VOC) profile in a murine model.

## 2. Results

### 2.1. Susceptibility Testing

(S)-reutericyclin exhibited antimicrobial activity against *Staphylococcus (S.) epidermidis* only. Neither cooking, buffering, nor treatment with HCl or proteinase K influenced this activity (Figure 1).

### 2.2. In Vivo Microbiome Analysis

The microbiome analysis revealed a trend towards reduced alpha diversity and significantly reduced Bray–Curtis dissimilarity in animals receiving (S)-reutericyclin (Figure 2).

The linear discriminant analysis effect size (LEfSe) analysis at the species level between the two groups showed significant differences in mice undergoing (S)-reutericyclin treatment compared to the control group (Figure 3).

### 2.3. Fecal Volatile Organic Compound Analysis

A total of 42 VOCs could be detected tentatively in the headspace of murine stool samples. The mean stool sample weight was 25 mg (SD 3.4 mg). The alterations of four substances (propene, isopropyl alcohol, isoflurane and o-xylene) could be attributed to room air contamination. Another five substances (propanol, acetic acid, 2-methylpentane, 3-methylpentane and butanoic acid butyl ester) were detected in <2 samples and were thus excluded from the further analysis. This left 33 candidate substances for statistical comparison. Pentane and heptane were significantly increased, and 2,3-butanedione and 2-heptanone were significantly decreased in animals of the (S)-reutericyclin group. Butanoic acid propyl ester showed a trend towards an increase and (z)-2-butene and acetoacetate methyl ester towards a decrease in mice gavage fed with (S)-reutericyclin. Dendrogram analysis revealed a clustering within ketones, esters and short chained carbohydrates (Figure 4).

### 2.4. Correlation Analysis

The results of the Spearman-Rho correlation analysis between the bacteria altered in LEfSe analysis and VOCs with a group difference *p* < 0.1 are shown in Figure 5.

## 3. Discussion

In this investigation we tested the in vitro antimicrobial activity of (S)-reutericyclin and found a different reactivity than expected from the (R)-isoform reported in the literature. Furthermore, we gavage fed mice with (S)-reutericyclin and detected a shift in the microbiome towards dysbiosis compared to control animals. Finally, (S)-reutericyclin lead to distinct alterations of the fecal VOC profile as a marker for the bacterial metabolism.

In our in vitro susceptibility testing (S)-reutericyclin exhibited antimicrobial effects against *Staphylococcus (S.) epidermidis*, but not against the other bacteria tested. While the lack of an effect on Gram-negative bacteria (*Escherichia (E.) coli*, *Campylobacter (C.) jejuni* and *Pseudomonas (P.) aeruginosa*) was expected, a reactivity against the other Gram-positive bacteria should have been detectable [15]. Our finding is in sharp contrast to examinations by Hurdle et al. who described excellent potency of reutericyclins ((R)-reutericyclin and chemical modifications) against the lethal non-growing stage of *Clostridium (C.) difficile* at concentrations that may also be attained in the gastrointestinal tract [17]. In their study, they describe a mean minimal inhibitory concentration (MIC) of 0.09–0.5 mg/L depending on the strain [17]. Although we applied (S)-reutericyclin at a concentration of 62.5 µg/mL (resulting in 1.25 µg/platelet for the 1:1 concentration), we could not achieve an inhibitory effect. One reason for this finding may lie in the different culture methods. Hurdle et al. used TY broth, while we cultured on plates with tryptic soy broth supplemented with sheep blood. As reutericyclin showed activity against *S. epidermidis*, it obviously migrated into the culture medium. However, it may be possible that the final concentration in the agar was too low to inhibit *C. difficile* in our experiments. Another reason may be the different *Clostridium* strains used by Hurdle et al. (*C. difficile* BAA-9689, BAA-1803 and BAA-1875 [17]) in comparison to our experiment (non-toxin producing *C. difficile* ATCC 700057). The third possible reason may be the type of reutericyclin used. Chemically, there are two isoforms ((5S)- and (5R)-reutericyclin) and numerous different versions of chemically modified reutericyclins, which have been described in the literature [16,17]. In this regard, Hurdle et al. investigated (R)-reutericyclin and its 867 and 1138 modifications [17]. Similarly, Cherian et al. described inhibitory effects against Gram-positive bacteria (*Enterococcus (En.) faecalis* ATCC 33186, *Streptococcus (Str.) pyogenes* ATCC 700294, *Str. pneumonia*, *Bacillus anthracis sterne*, *Bacillus subtilis* ATCC 23857, *C. difficile* BAA 1803 and methicillin-susceptible *S. aureus* N315) depending on different modifications of the reutericyclin side chains [16]. It may be possible that the type of isoform ((R) or (S)) affects the antimicrobial activity of reutericyclin. As such, it might be possible that the (S)-isoform of reutericyclin used in this experiment exhibits a reduced antimicrobial activity compared to an (S)/(R) racemate, the (R)-isoform or other chemically modified versions [16,17] of reutericyclin.

Buffering, treatment with HCl or proteinase K or cooking did not influence the activity of (S)-reutericyclin in our study. While lacking effects of buffering or treatment with HCl or proteinase K are similar to results reported by Messens et al., we could not detect an influence of heat treatment in contrast to their study [14].

Regarding the in vivo application, there are currently no reports regarding the effect of either form of reutericyclin as a pure substance in mice. Despite a possible mitigation of the antimicrobial activity of reutericyclin by cecal contents, effective concentrations should be achievable in the colon [17,22]. While there is no such study on reutericyclin as pure substance, there are a few reports investigating the effect of the reutericyclin producing probiotic *L. reuteri* in piglets [20,21,23]. While these animal studies found no effect of reutericyclin on bacterial diversity we encountered a significant decrease of the Bray–Curtis dissimilarity and a trend towards reduced alpha diversity when applying (S)-reutericyclin directly as a postbiotic.

In piglets, administration of the reutericyclin-positive *L. reuteri* TMW1.656 transiently decreased the abundance of the *L. reuteri* group in fecal samples in comparison to controls and reduced the proportion of lactobacilli in comparison to the reutericyclin-negative mutant [21]. These results suggest that reutericyclin is a subtle but significant modulator of the Lactobacillus community in pigs [21]. In this regard we could demonstrate a tendency towards reduced levels of *L. murinus* in the (S)-reutericyclin group. In contrast, another study revealed different results regarding bacterial abundance. The authors showed that feed fermentation with *L. reuteri* affected the abundance of few bacterial taxa and particularly reduced the abundance of *Enterobacteriaceae* when compared to unfermented controls [23]. At the same time, reutericyclin-producing *L. reuteri* were found to increase the abundance of *Dialister* spp. and *Mitsuokella* spp. but did not influence the abundance of clostridial toxins in the feces [23]. Finally, Wang et al. reported that the probiotic-derived antimicrobial compound reutericyclin had only a limited impact on swine intestinal microbiota [20]. All of these investigations deal with probiotic administration of reutericyclin-producing *L. reuteri* and not with the pure substance itself. At present, it remains unclear if these bacteria produce the (R)-form, (S)-form or a racemate.

While probiotic derived reutericyclin seemed to have a minor effect in swine, mice reacted to (S)-reutericyclin as pure substance with a reduction of potentially beneficial butyrate producers (*Roseburia*, *Eubacterium* and *Ruminococcus*) [24]. Additionally, the increase of *Streptococcus* has also been encountered in humans—for instance under therapy with proton pump inhibitors [25]. Amongst others, an increase of *Streptococcus* in the fecal microbiome has been associated with an increased risk of a *Clostridium difficile* infection in humans [26].

Studies concerning *Muribaculum* are restricted to a mouse model reporting a decreased relative abundance of *Muribaculum* in mice developing inflammatory bowel disease [27]. Thus, *Muribaculum* (amongst others) may be linked to inflammatory processes localized in the intestinal wall. Overall, the application of (S)-reutericyclin caused a decrease of potentially beneficial and an increase of potentially harmful bacteria in our mouse model.

Volatile organic compounds in the headspace of fecal samples are generated during bacterial metabolic processes [28]. VOCs are responsible for the “smell” of different microbial species [29]. Consequently, cultures of some bacteria such as *Escherichia coli*, *Mycobacterium* spp., or molds like *Aspergillus* und *Fusarium* spp. have distinct VOC profiles allowing recognition of species by their “smell” [30,31,32,33]. VOCs within the gut are influenced by intestinal epithelium, the microbiome and diet [28]. In mammals, usually esters dominate the fecal VOC profile as also seen in our measurements. The short carbohydrates pentane and heptane, which were increased under (S)-reutericyclin treatment are substances commonly found in polluted air [34,35]. 2,3-butanedione was decreased in (S)-reutericyclin fed mice. It is a plant-growth promoting compound [36] emitted from *Bacillus subtilis* [37] and has been reported in the headspace of *Pseudomonas pseudoalcaligenes* [38] and *Bacillus mojavensis* [36] cultures, but was not previously reported in fecal samples of mice. Among other substances 2-heptanone showed elevations in tissue and fecal samples of dairy cattle and goats infected with *Mycobacterium avium* subspecies *paratuberculosis* [39]. In an urinalysis of healthy adult humans, 2-heptanone could be detected among many other VOCs [40]. Furthermore, 2-heptanone was found to decrease in humans subjected to altitude induced hypoxia [41]. In our study, VOC measurements correlated to fecal bacterial abundances to some extent. However, the intestinal microbiome is highly variable throughout the gastrointestinal tract [42] and volatile metabolites are not exclusively produced by certain species. As such, it is almost impossible to directly relate certain VOC substances to specific bacteria in vivo.

One possible limitation of our study is that we only tested one form of reutericyclin ((S)-reutericyclin). Regarding susceptibility testing, this can be justified by the wide variety of reports available for (R)-reutericyclin and its modifications. Additionally, we did not test the susceptibility against other *Lactobacillus species*. In future studies we plan to elucidate the effect of the different isoforms on the antimicrobial activity against various bacteria including *Lactobacillus species* of reutericyclin.

In a previous investigation we could demonstrate the variability of the intestinal microbiome throughout the gut, even showing differences between luminal and mucosal microbial compositions [42]. It might well be possible, that animals showed relevant alterations at other intestinal levels. However, in this study we limited our microbiome analysis to fecal samples only.

Before initiating this study, it was unclear if the isoform would have an effect on the fecal microbiome of mice at all. It was ethically impossible to include various isoforms and modifications in an animal study without evidence of such an effect. Giving the findings of the present investigation, this is another issue that remains to be addressed in the future.

In conclusion, (S)-reutericyclin had an inhibitory effect on *S. epidermidis* but exhibited a different antimicrobial activity than described for the (R)-isoform. Moreover, its application in a murine model caused microbial alterations towards dysbiosis.

## 4. Materials and Methods

### 4.1. Susceptibility Testing

Bacteria for resistance testing were obtained from Aurosan GmbH, Essen, Germany. *C. difficile* (ATCC 700057), *Listeria (Lis.) monocytogenes* (ATCC 15313), *E. coli* (ATCC 25922), *En. faecium* (ATCC 27270), *S. aureus* (ATCC 29213), *S. epidermidis* (ATCC 12228), *Str. agalactiae* (ATCC 13813), *P. aeruginosa* (ATCC 27853), and *Propionibacterium (Pr.) acnes* (ATCC 6919) were chosen for bacterial resistance testing. After thawing, bacteria were pre-cultured as broth cultures. Aerobic bacteria were cultivated for 24 h at 37 °C and 120 rpm and anaerobic bacteria for 48 h at 37 °C and 120 rpm. Of each pre-culture, 300 µL were then spread on 15 cm diameter plates with either tryptic soy agar (30 g/L TSB, Fluka Analytical, no T8907-500G, Honeywell, Charlotte, NC, USA; 15 g/L Agar-Agar, Kobe I, no5210-2, Carl Roth GmbH, Karlsruhe, Germany; TSA) for *E. coli*, *S. aureus* and *P. aeruginosa*, TSA with 5% freshly harvested sheep blood for *C. difficile*, *Str. agalactiae* and *Pr. acnes*, brain heart infusion agar (Brain-Heart-Infusion, no X915.1, 52 g/L, Carl Roth GmbH, Karlsruhe, Germany; 15 g/L Agar-Agar Kobe I, no5210-2, Carl Roth GmbH, Karlsruhe, Germany) for *En. faecium* and *Lis. monocytogenes* or nutrient agar (1 g/L beef extract powder, no B4888-50G, Sigma-Aldrich Handels GmbH, Vienna, Austria; 5 g/L peptone, no P0431-250G, Sigma-Aldrich, Handels GmbH, Vienna, Austria; 5 g/L NaCl and 15 or 50 g/L Agar-Agar) for *S. epidermidis*. Anaerobic bacteria (*C. difficile* and *Pr. acnes*) were cultured for 48 h in boxes with oxid Anaerogen 2.5 L (Thermo Fisher Scientific, Waldham, MA, USA). The remaining aerobic bacteria grew for 24 h until a dense bacterial lawn was achieved.

Each microorganism was cultured on five different plates. Using a stencil, 9 disks for resistance testing (BD Sensi-Disc^TM^, Becton, Dickinson and Company, Franklin Lakes, NJ, USA) were placed on each culture plate using a prepared scheme. (S)-reutericyclin was purchased from BioCrick Biotech Co., Ltd. (Sichuan, China). (S)-reutericyclin was dissolved at a concentration of 62.5 µg/mL in sterile water containing 1‰ H_3_PO_4_. Each disk was either treated with respectively 20 µL of 1:1 (S)-reutericyclin solution, a 1:2 or 1:4 dilution of the solution, cooked solution (100 °C for 30 min, no rotation, Thermomixer, HLC, Germany), buffered solution (to pH 7 with 1 n NaOH), solution mixed with 1 n HCl (1:1) or solution treated with 1 mg/mL Proteinase K (Carl Roth, Germany). Sterile water with 1‰ H_3_PO_4_ served as negative control. Either vancomycin (Vancomycin Hikma^®^ 500 mg, Hikma Pharma, Planegg, Germany; 0.03 mg/disk) for *C. difficile*, *Str. agalactiae*, *En. faecium*, *S. epidermidis*, *Lis. monocytogenes* and *S. aureus*, or Piperacillin/Tacobactam (PIPeracillin/TAZobactam Kabi 4 g/0.5 g, Fresenius Kabi, Graz, Austria; 0.1 mg/disk) for *E. coli*, *P. aeruginosa* and *Pr. acnes* were used as positive controls. Plates were then incubated at 37 °C for 24 h in the case of aerobic and for 48 h in the case of anaerobic bacteria. Thereafter, plates were photographed and inhibition zones were determined with ImageJ 2.0.0-rc-69/1.52p (ImageJ open source image processing software, http://imagej.net/Contributors, accessed on 12 December 2020).

### 4.2. Animal Model

Male BALBc mice (*n* = 20) were obtained at an age of 7 weeks from the Center for Biomedical Research of the Medical University of Vienna, Austria, as one batch of littermates for microbiome testing. After delivery and an acclimatization period of two weeks, mice were split, forming two equal groups (*n* = 10 each) with equal body weight distribution. Mice were kept single-housed in individually ventilated cages under specific pathogen free conditions, a 12 h light-dark cycle and free access to chow and water at all times. After acclimatization, mice underwent a daily gavage with (S)-reutericyclin as follows: 12.5 µg/mouse/day in 400 µL sterile water containing 1‰ H_3_PO_4_ in the first week to check if mice accepted (S)-reutericyclin without complications. In weeks 2–5, the dosage was increased to 25 µg/mouse/day in 400 µL sterile water containing 1‰ H_3_PO_4_ and in weeks 6–7 to 50 µg/mouse/day in 400 µL sterile water containing 1‰ H_3_PO_4_ (*n* = 10 mice). Sterile water containing 1‰ H_3_PO_4_ served as control (*n* = 10 mice). After 7 weeks of gavage, mice were euthanized by cranio-cervical dislocation. Two stool samples were collected on the day of euthanasia. One was stored at −80 °C until 16S based microbiome analysis. The other sample was immediately sent for VOC analysis.

### 4.3. 16S Based Microbiome Analysis

For total DNA isolation, fecal samples were isolated with the Magna Pure LC DNA III Isolation Kit (Bacteria, Fungi) (Roche, Mannheim, Germany) according to published protocols [43]. Briefly, one stool pellet was mixed with 500 µL PBS and 250 µL bacterial lysis buffer. Samples were homogenized and bead beaten in Magna Lyzer Green bead Tubes (Roche, Mannheim, Germany) in a Magna Lyzer instrument (Roche, Mannheim, Germany) at 6500 rpm for 30 s two times. Followed by enzymatic lysis with 25 µL lysozyme (100 ng/mL, 37 °C for 30 min) and 43.4 µL proteinase K (20 mg/mL, 65 °C for 1 h) samples were heat inactivated at 95 °C for 10 min and total DNA was purified in a MagnaPure LC instrument (Roche, Mannheim, Germany) according to manufacturer’s instructions. Total DNA was eluted in 100 µl elution buffer and stored at −20 °C until analysis. For 16S PCR, 2 µL of total DNA were used as a template in a 25 µL PCR reaction with the FastStart™ (Sigma Aldrich Handels GesmbH, Vienna, Austria) High Fidelity PCR-System (Sigma, Darmstadt, Germany) according to the manufacturer’s instructions and the target specific primers 515F (5′-GTGYCAGCMGCCGCGGTAA-3′) and 806R (5′-GGACTACNVGGGTWTCTAAT-3′) for 30 cycles in triplicates. Triplicates were pooled, normalized, indexed and purified according to published protocols [43]. The final pool was sequenced on an Illumina MiSeq desktop sequencer at 9 pM and v 3 600 cycles chemistry. FASTQ raw files were used for data analysis.

A total of 2,479,083 MiSeq paired end FASTQ reads were used for further analysis. The DADA2 pipeline for modeling and correcting Illumina-sequenced amplicon errors for quality-filtering [44] was used with standard settings for denoising, dereplicating, merging and check for chimeras as implemented in QIIME2 2018.4 microbiome bioinformatics platform [45]. QIIME2 was integrated in our own non-public instance of Galaxy (MedBioNode https://galaxy.medunigraz.at accessed on 12 August 2020) [46]. Taxonomic assignment of the DADA2 representative sequences was provided with the QIIME2 sklearn-based classifier against SILVA rRNA database release 132 at 99% identity [47]. To interpret and compare taxonomic information, 16S rRNA data was transferred to the Calypso online software (Calypso 8.84^®^, accessible through http://cgenome.net/wiki/index.php/Calypso, last accessed 24 August 2020) [48]. Samples were rarefied to a read depth of 12,774. Alpha diversity was calculated using Chao1 estimator, Inversed Simpson and Shannon index. Relative abundances (total sum scaling with square root transformation) were used for further group comparisons. Beta diversity was examined using a redundancy analysis (RDA) and colored principal component analysis plots (PCoA) based on Bray–Curtis dissimilarity score. The identification of discriminating taxa between the groups was performed with a linear discriminant effect size (LEfSe) analysis. Differentially abundant taxa identified by LEfSe analysis were considered relevant if the differences between groups could be verified by ANOVA (*p* < 0.1).

### 4.4. Stool VOC Analysis

Samples were weighed and stored in glass vials (Gerstel GmbH, Germany) and stored at 6 °C. Room air samples were collected at the same time points to correct for possible contamination. All samples were immediately sent to the partner via overnight express for gas chromatography/mass spectrometry (GC-MS). VOC analysis was performed in the headspace of samples as previously reported [49,50,51]. VOCs were pre-concentrated with a commercially available solid phase micro extraction (SPME) fiber (carboxen/polymethylsiloxane, Supelco, Bellefonte, PA, USA). An Agilent 7890 A gas chromatograph (GC) coupled to an Agilent 5975 C inert XL mass selective detector (MSD) was used to separate and identify the VOCs desorbed from the SPME device. Detected marker substances were identified tentatively from a mass spectral library (National Institute of Standards and Technology 2005; NIST 2005, Gatesburg, PA, USA) and by retention time matching. Results were corrected for the stool weight. When the mean of the room air samples exceeded 30% of the mean of the headspace samples, a possible contamination was recorded and the substance was excluded from further analysis. The responses of a selected m/q ratio at a defined retention time for each substance were recorded, integrated and used for group comparison.

### 4.5. Statistics

Data was managed with Microsoft Excel 2016^®^. For statistical analysis, data was transferred to SPSS 26.0^®^. Graphical work-up was performed with GraphPad Prism 9^®^. The Heatmap was drawn with the heatmap function of ggplot2 package (version 3.3.3) for RStudio^®^ (version 1.4.1106). Metric data is displayed as median and interquartile range (IQR). A Mann–Whitney U-Test was used to determine group differences. Correlation analyses were conducted between bacteria resulting from LEfSe analysis and VOCS with group differences *p* < 0.1 using the corrplot package (version 0.84) for RStudio^®^ applying a Spearman-Rho Test. *P*-values < 0.05 were considered statistically significant.

## Figures and Tables

**Figure 1 ijms-22-06424-f001:**
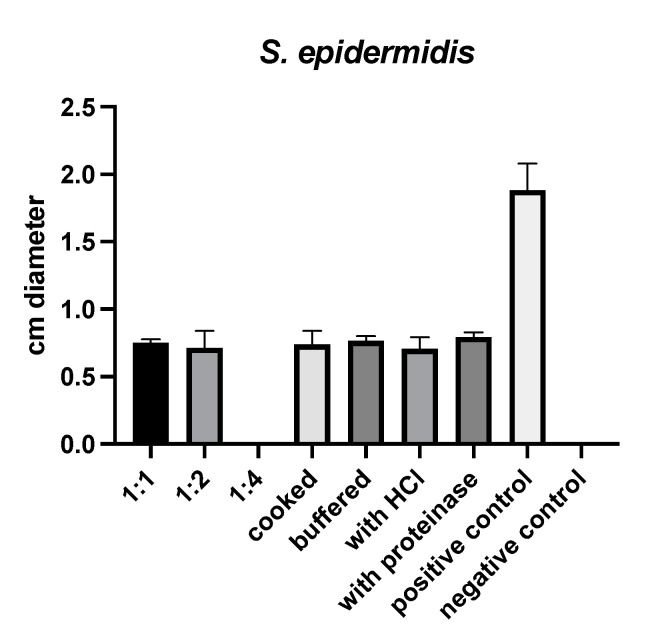
Results of susceptibility testing with (S)-reutericyclin.

**Figure 2 ijms-22-06424-f002:**
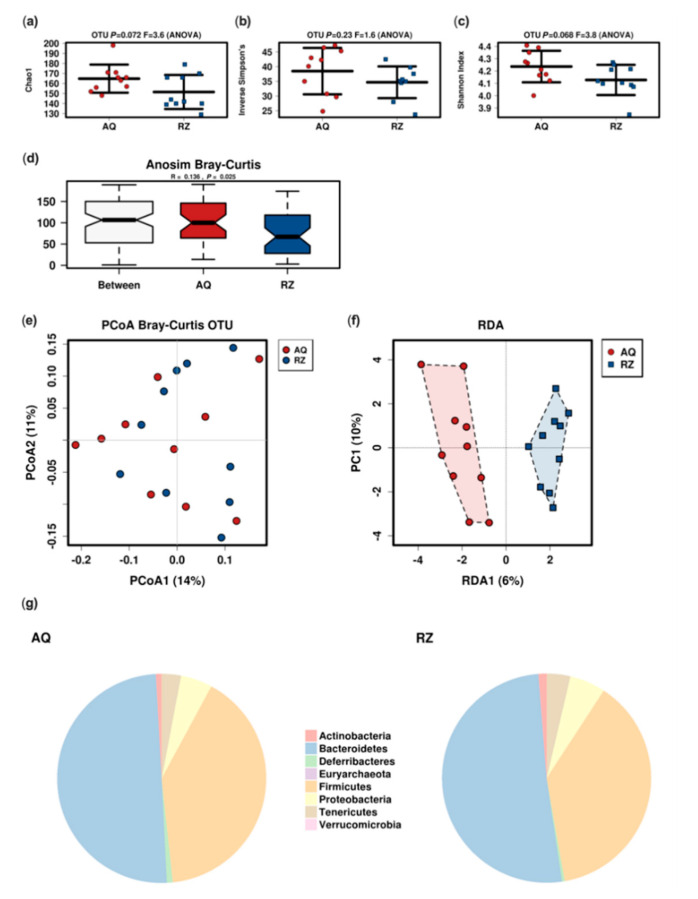
Microbiome analysis of the two groups. Alpha (**a**–**c**) and beta-diversity; (**d**–**f**) indices (RDA: variance 23.12; F 1.17; *p* 0.07); pie charts of the mean relative abundance at the phylum level (**g**). AQ…aqua control group; RZ…(S)-reutericyclin group.

**Figure 3 ijms-22-06424-f003:**
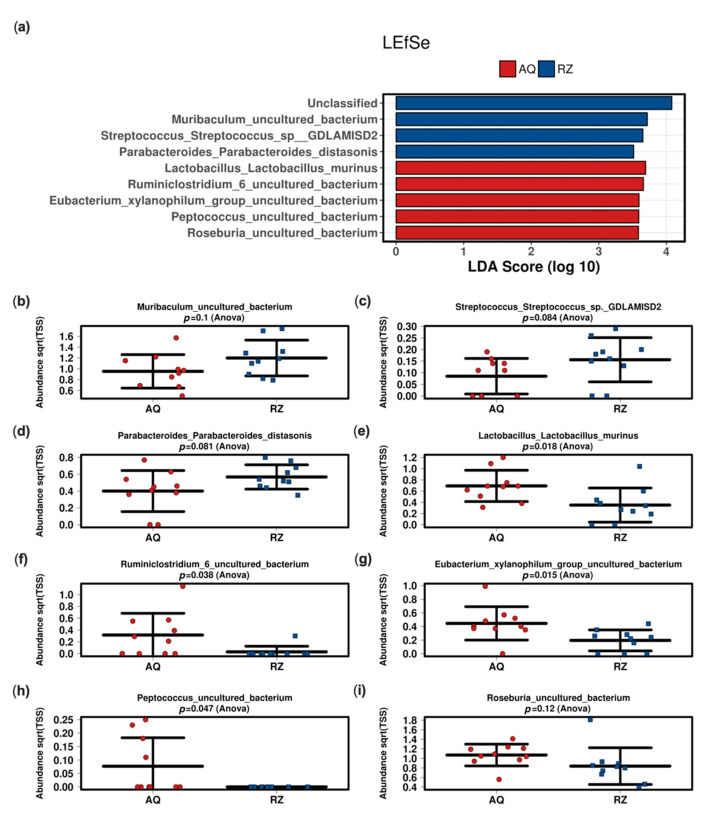
Linear discriminant (LDA) and effect size (LEfSe) analysis of the two groups (**a**). Strip charts (mean and standard deviation) of the bacteria altered in the LEfSe analysis (**b**–**i**). AQ—aqua control group; RZ—(S)-reutericyclin group.

**Figure 4 ijms-22-06424-f004:**
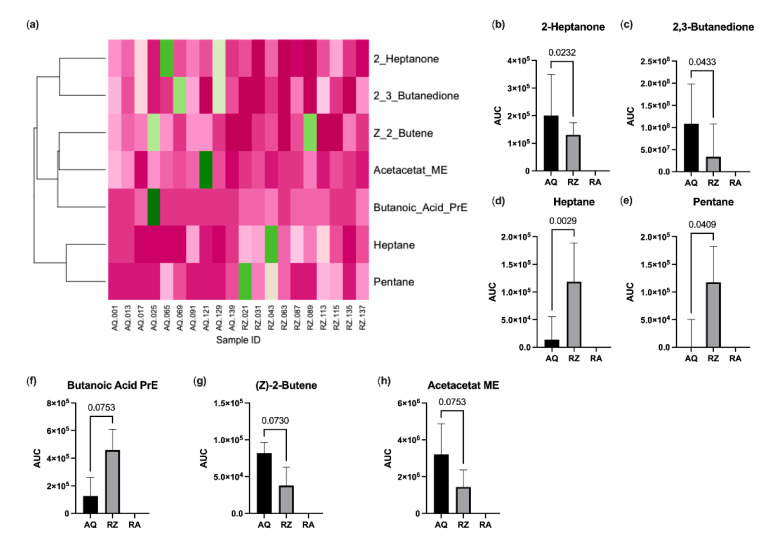
Results of the VOC analysis. (**a**) Heatmap with dendrogram. VOC values were normalized to the maximum; colors range from dark green (=1.0) to dark pink (=0.0); substances with significant changes (**b**–**e**) and with a trend between (**f**–**h**) the groups. AQ…aqua control group; RZ…(S)-reutericyclin group; RA…room air samples; PrE…propyl ester; ME…methyl ester.

**Figure 5 ijms-22-06424-f005:**
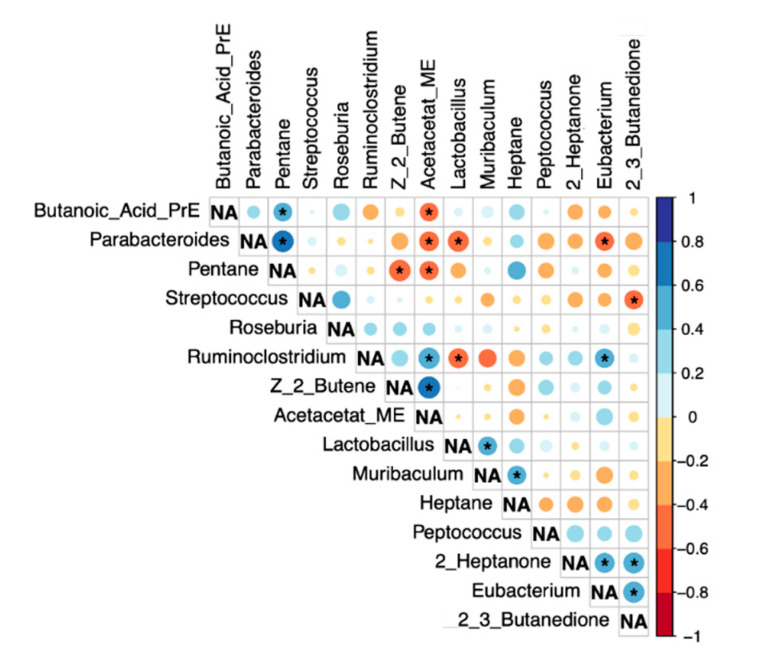
Correlation analysis (Spearman-Rho). Significant correlations (*p* < 0.05) are marked with *. NA…not applicable.

## Data Availability

Data from microbiome and VOC analysis is available from a Mendeley Data^®^ repository (access: http://dx.doi.org/10.17632/xzb8wzxt5z.1).

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
