# Peer review of "(S)-Reutericyclin: Susceptibility Testing and In Vivo Effect on Murine Fecal Microbiome and Volatile Organic Compounds"

_ijms, 2021, doi:10.3390/ijms22126424_

Round 1

Reviewer 1 Report

Dear authors,

The subject of your manuscript entitled "(S)-Reutericyclin: Susceptibility Testing and in vivo Effect on Murine Fecal Microbiome and Volatile Organic Compounds" submitted for publication in the International Journal of Molecular Sciences falls within the scope of this journal. The manuscript is well-written and it has scientific soundness.

I have only one observation:

In lines 78-79 you mentioned that (S)-reutericyclin exhibited antimicrobial activity against S. epidermidis only. Neither cooking, buffering, nor treatment with HCl or proteinase K influenced this activity (Figure 1). Then, in the Discussion section, lines 159-160 you say the opposite: Buffering, treatment with HCl or proteinase K or cooking influenced the activity of (S)-reutericyclin in our study. This is in accordance with previous reports, where inhibitory substances produced by lactobacilli were isolated from sourdough [14]. Please explain and correct accordingly.

Author Response

REVIEWER: In lines 78-79 you mentioned that (S)-reutericyclin exhibited antimicrobial activity against S. epidermidis only. Neither cooking, buffering, nor treatment with HCl or proteinase K influenced this activity (Figure 1). Then, in the Discussion section, lines 159-160 you say the opposite: Buffering, treatment with HCl or proteinase K or cooking influenced the activity of (S)-reutericyclin in our study. This is in accordance with previous reports, where inhibitory substances produced by lactobacilli were isolated from sourdough [14]. Please explain and correct accordingly.

RESPONSE: Thank you very much for this remark. We apologize for this error. we corrected the discussion section accordingly. It now reads "Buffering, treatment with HCl or proteinase K or cooking did not influence the activity of (S)-reutericyclin in our study. While lacking effects of buffering or treatment with HCl or proteinase K are similar to results reported by Messens et al we could detect no influence of heat treatment in contrast to their study [14]."

Reviewer 2 Report

The aims of this study were to examine 1) the in vitro antimicrobial activity of (S)- reutericyclin and 2) its in vivo effect on the fecal microbiome and 3) fecal volatile organic compound (VOC) profile in a murine model.

Introduction is written very well but I miss some information about animal microbiota influenced by reutericyclin.

Materials and methods are described as methods of antimicrobial activity and antibiotic resistance as control. What I miss there is evaluation of different concentrations of reutericyclin. It is very important to know minimal inhibitory concentration. For animal models, ethical statements are missing.

Results: How do authors explain the fact that reutericycline inhibited only S. epidermidis? In discussion more sentences about the microbiota of mouse isolation are missing.

In all manuscripts there is a necessary conclusion as a separate part. Short conclusion is part of the discussion. This part must be described.

Author Response

REVIEWER: Introduction is written very well but I miss some information about animal microbiota influenced by reutericyclin.

RESPONSE: Thank you very much for this remark. In our first version we included this information in the discussion section of the manuscript. We have now included a short paragraph on the effect of reutericyclin on the fecal microbiome in the introduction section.

REVIEWER: Materials and methods are described as methods of antimicrobial activity and antibiotic resistance as control. What I miss there is evaluation of different concentrations of reutericyclin. It is very important to know minimal inhibitory concentration. For animal models, ethical statements are missing.

RESPONSE: Thank you very much for this comment. As stated in the text of materials and methods and also displayed in figure 1 we have tested different dilutions of reutericyclin. 1:1 contained 12.5µg/20µl; 1:2 6.25µg/µl and 1:4 3.125µg/µl. Please compare figure 1.

REVIEWER: Results: How do authors explain the fact that reutericycline inhibited only S. epidermidis? In discussion more sentences about the microbiota of mouse isolation are missing.

RESPONSE: As stated in the discussion section there seem to be different possible explanations: "Although we applied (S)-reutericyclin at a concentration of 62.5 µg/ml (resulting in 1.25µg/platelet for the 1:1 concentration) we could not achieve an inhibitory effect. One reason for this finding may lie in the different culture methods. Hurdle et al used TY broth while we cultured on plates with tryptic soy broth supplemented with sheep blood. As reutericyclin showed activity against S. epidermidis it obviously migrated into the culture medium. However, it may be possible that the final concentration in the agar was too low to inhibit C. difficile in our experiments. Another reason may be the different Clostridium strains used by Hurdle et al (C. difficile BAA-9689, BAA-1803 and BAA-1875 [17] in comparison to our experiment (non-toxin producing C. difficile ATCC 700057). The third possible reason may be the type of reutericyclin used. Chemically, there are two isoforms (5S)- and (5R)-reutericyclin and numerous different versions of chemically modified reutericyclins which have been described in the literature [16,17]. In this regard Hurdle et al. investigated (R)-reutericyclin and its modifications 867 and 1138 [17]. Similarly, Cherian et al. describe inhibitory effects against Gram-positive bacteria (Enterococcus (En.) faecalis ATCC 33186, Streptococcus (Str.) pyogenes ATCC 700294, Str. pneumonia, Bacillus anthracis sterne, Bacillus subtilis ATCC 23857, C. difficile BAA 1803 and methicillin-susceptible S. aureus N315) depending on different modifications of the reutericyclin side chains [16]. It may be possible that the type of isoform ((R) or (S)) affects the antimicrobial activity of reutericyclin. As such, it might be possible, that the (S)-isoform of reutericyclin used in this experiment exhibits a reduced antimicrobial activity compared to an (S)/(R) racemate, the (R)-isoform or other chemically modified versions [16,17] of reutericyclin." For the isolation of the microbiota from fecal samples we allow to refer to our methods section.

REVIEWER: In all manuscripts there is a necessary conclusion as a separate part. Short conclusion is part of the discussion. This part must be described.

RESPONSE: Thank you very much for this comment. In this regard please allow us to refer to our conclusion at the end of the discussion section reading "In conclusion, (S)-reutericyclin had an inhibitory effect on S. epidermidis but exhibited a different antimicrobial activity than described for the (R)-isoform. Moreover, its application in a murine model caused microbial alterations towards dysbiosis."

Reviewer 3 Report

This study aimed to assess the in vitro antimicrobial activity and the in vivo effect of (S)-reutericyclin on the murine fecal microbiome and volatile organic compound (VOC) profile. This is the study that looked at the reutericyclin isoform in vivo and in vitro effect and correlated findings to change in the VOC profile. Below are the reviewers’ comments;

Line 77, since this was the first study to investigate the (S)-reutericyclin effects, authors should test (S)-reutericyclin susceptibility against other relevant lactobacillus species and report the findings accordingly in the manuscript.

Figure 5, improve the labeling fonts and characters. It is hard to appreciate the labeling.

Lines 158-161, it should be different treatment did not influence the effect. Correct this.  

Lines 283-285, since (S)-reutericyclin dose was periodically increased with the control group being constant how do you explain the observed findings then?  

If animals were euthanized at week-7, why did the authors investigate fecal samples instead of intestinal samples? Ideal small intestinal and large intestinal contents can be investigated and compared with and between-group. Explain

Author Response

REVIEWER: Line 77, since this was the first study to investigate the (S)-reutericyclin effects, authors should test (S)-reutericyclin susceptibility against other relevant lactobacillus species and report the findings accordingly in the manuscript.

RESPONSE: Thank you very much for this remark. We completely agree to the reviewer, that resistance testing of Lactobacillus species would be interesting. However in this study we chose to focus on other bacteria related to an investigation of Cherian et al 2014. Furthermore, the Journal only allowed 7 days for the revisions and it is impossible to order, cultivate and test bacteria in this time. We have included a remark regarding tests of Lactobacillus species in the study limitations section and plan to address this issue in a future investigation. 

REVIEWER: Figure 5, improve the labeling fonts and characters. It is hard to appreciate the labeling.

RESPONSE: We agree, that the text was a little small and blurry. We have included an improved version of the figure in the revised manuscript now.

REVIEWER: Lines 158-161, it should be different treatment did not influence the effect. Correct this.  

RESPONSE: Thank you very much for this comment. We apologize for this error and have revised the discussion section accordingly.

REVIEWER: Lines 283-285, since (S)-reutericyclin dose was periodically increased with the control group being constant how do you explain the observed findings then?  

RESPONSE: Thank you for this remark. Our mice were always gavage fed with 400µl. The control group received autoclaved water with 1‰ H3PO4. The study group received increasing dosages of reutericyclin in 400µl autoclaved water with 1‰ H3PO4 per gavage. As such the volume per gavage remained constant but the dosage of reutericyclin increased (from 12.5µg/day to 50µg/day).

REVIEWER: If animals were euthanized at week-7, why did the authors investigate fecal samples instead of intestinal samples? Ideal small intestinal and large intestinal contents can be investigated and compared with and between-group. Explain

RESPONSE: Thank you very much for this comment. We chose to investigate the fecal microbiome as also reported in piglets by other authors. Indeed we could demonstrate a big variation of the intestinal microbial composition throughout the intestine and even between mucosal and luminal locations (Klymiuk et al 2021; PMID 33806771). As such the microbiome varies within the mouse. In the present setting of this study however it was impossible to investigate all of these different levels. We have included a section regarding this problem in the study limitations section.

Round 2

Reviewer 2 Report

All comments were accepted.

Reviewer 3 Report

none